# Human Evidence of Perfluorooctanoic Acid (PFOA) Exposure on Hepatic Disease: A Systematic Review and Meta-Analysis

**DOI:** 10.3390/ijerph191811318

**Published:** 2022-09-08

**Authors:** Jihee Choi, Jong-Yeon Kim, Hae-Jeung Lee

**Affiliations:** 1Department of Food and Nutrition, College of Bionanotechnology, Gachon University, Seongnam-si 13120, Gyeonggi-do, Korea; 2Institute for Aging and Clinical Nutrition Research, Gachon University, Seongnam-si 13120, Gyeonggi-do, Korea; 3Department of Food Science and Biotechnology, Gachon University, Seongnam-si 13120, Gyeonggi-do, Korea

**Keywords:** perfluorooctanoic acid, hepatic diseases, liver, systematic review, meta-analysis

## Abstract

Background: Perfluorooctanoic acid (PFOA) is widely used throughout different industries, including the food industry, because it is resistant to heat and prevents water or oil from easily permeating into or contaminating materials coated by PFOA. Although many studies have reported an association between PFOA exposure and the risk of developing hepatic diseases, it is still in debate because they have shown conflicting results. Therefore, this study conducted a systematic review and meta-analysis on the relationship between PFOA exposure and hepatic diseases. Methods: This study searched studies related to hepatic diseases due to PFOA exposure until 31 December 2021, using PubMed, EMBASE, and Web of Science. This study performed a systematic review and meta-analysis through research question development, literature screening, data extraction, and risk of bias evaluation. This study found 8280 studies after excluding duplicate literature and selected 5 studies in the final stage. Among them, two studies were included in the meta-analysis. Results: The results of the meta-analysis showed that the ALT of people exposed to PFOA was 117% higher than the ALT of those not exposed to PFOA, and it was significantly different (OR = 1.167; 95% CI, 1.086–1.254). Conclusion: However, since the number of studies included in the analysis was not large enough to conclude that PFOA exposure was associated with the development of hepatic diseases, more observational studies are needed to confirm its long-term effects.

## 1. Introduction

Perfluorooctanoic acid (PFOA) belongs to a family of perfluorinated compounds (PFCs) with strong carbon and fluorine (C-F) bonds and is the most stable and detectable among the PFCs [1]. Due to two major characteristics, hydrophilicity and hydrophobicity, PFOA has been used for industrial and consumer products as a coating on food packaging materials, non-stick cookware, and furniture [2,3]. From 1951 to 2004, the total global production of PFOA was estimated at 3600–5700 t, and the total global emission of PFOA was estimated at 400–700 t. Released PFOA has been found in the air, water, and soil, and it is of great concern that it may accumulate in the ecosystem and organisms [4]. Thus, PFOA is a potential hazard and requires caution when using it.

PFOA exposure may cause various toxicities to humans and animals due to the long half-life of PFOA. In humans, PFOA exposure can lead to liver, spleen, and testicle cancers, as well as reproductive toxicity [5]. Moreover, it can result in hepatic, cardiovascular, and reproductive toxicities, as well as disorders in the immune and endocrine system of animals [6]. In particular, a recent study revealed that PFOA could cause fatty liver in adults and non-alcoholic fatty liver in children [7]. In addition, when pregnant Kunming mice were exposed to PFOA, it reduced the growth and development of the pups and induced liver damage, which disrupted the secretion of enzymes involved in PPAR-α-induced fatty acid oxidation, which resulted in hepatic bleeding, local necrosis, enlargement of hepatocytes, and decrease in histone acetylation [8].

The production and usage of PFOA have gradually decreased in recent years as the risks with PFOA have surfaced. Furthermore, PFOA was recognized as an updated persistent organic pollutant (POP) to the Stockholm Convention on Persistent Organic Pollutants in 2020 [9]. However, PFOA is still used, mainly in developing countries in Eastern Europe and in Western European countries [10].

This systematic review study focused on the effects of PFOA exposure on human health, such as hepatic toxicity, and showed the human epidemiologic evidence related to PFOA exposure.

## 2. Materials and Methods

This systematic review was conducted according to the U.S. Environmental Protection Agency (EPA) systematic review protocol for perfluoroalkyl substances (PFASs), including epidemiology studies [11], and the Preferred Reporting Items for Systematic Reviews and Meta-Analyses (PRISMA) guidelines.

### 2.1. Specifying the Research Question

We developed research questions based on the PECO (Populations, Exposures, Comparators, and Outcomes) criteria to screen the PFOA-relevant studies. Our PECO criteria are as follows:Population: Any population and life stage (occupational or general population, including children and other sensitive populations). The controlled exposure, cohort, or cross-sectional studies were used.Exposure: Studies providing quantitative estimates of PFOA exposure based on administered dose or concentration, biomonitoring data (e.g., urine, blood, or other specimens), and environmental or occupational setting measures.Comparator: A comparison or reference population exposed to a lower level (0 or no exposure/exposure below detection levels) or for a shorter period.Outcome: All hepatic disease types.

### 2.2. Literature Search and Screening

A literature search was systematically performed using online databases (i.e., PubMed (MEDLINE), EMBASE, and Web of Science) without restricting language until December 2021. The search keywords included the following Medical Subject Headings (MeSH) or EMTREE terms: “perfluorooctanoic acid”, “PFOA”, “FC 143”, “335-67-1”, and “hepatic”. The complete search strategy is presented (Appendix A). The literature eligibility criteria are as follows (Appendix A).

Literature Selection Criteria: We selected studies associated with all hepatic disease types in which human exposure to PFOA was measured or estimated.Literature Exclusion Criteria: We excluded studies if they did not contain original data or were epidemiology studies (i.e., prospective cohort studies, nested case-control studies, and case-cohort studies), if study subjects were not humans, if the study subject’s exposure to PFOA was not measured or estimated, or if they did not provide outcome/exposure of interest.Study Selection Process: We excluded duplicates that were found in two or more databases. Abstract screening was performed by using Rayyan, open-source, online software, by two investigators independently [12]. Two independent researchers examined the abstracts and titles of all citations using the defined criteria. Studies that were not excluded based on the title and abstract were screened through a full text review. After the initial screening, when there was a discrepancy between the researchers, they discussed each discrepancy and brought in the third researcher, if necessary, to discuss and decide whether to include or exclude each discrepancy.

### 2.3. Data Extraction and Rating the Risk of Bias (RoB) 

Two researchers independently extracted data (e.g., study characteristics and results) from each selected study by using the EPA’s standardized data extraction domain [11]. Discrepancies were resolved by a discussion of two researchers. To assess the RoB, we used the validated EPA domain of evaluation for epidemiology studies [11], including all studies. The RoB was defined as the characteristics of a study that could introduce a systematic error in the magnitude or direction of the results [13]. The EPA domain of evaluation for epidemiology studies is partially modified from Cochrane’s Risk of Bias in Non-randomized Studies of Interventions (ROBINS-I). The EPA domain of evaluation for epidemiology studies consists of seven evaluation items: (1) exposure measurement, (2) outcome measurement, (3) participant selection, (4) potential confounding, (5) analysis, (6) selective reporting, and (7) sensitivity. We rated each item of the EPA domain of evaluation for epidemiology studies as “good”, “adequate”, “deficient”, or “critically deficient”. Two researchers independently judged RoB for each study across all domains.

### 2.4. Quantitative Synthesis and Meta-Analysis

The I^2^ (%) statistic was used to evaluate the heterogeneity of the selected studies. In general, the I^2^ (%) statistic is interpreted as low (I^2^ = 25%), intermediate (I^2^ = 50%), and high (I^2^ = 75%) [14]. The meta-analysis assumes that individual studies are independent. However, when one study reported multiple effect sizes, the independence assumption could be violated because the data used in the study might be used repeatedly. Therefore, this study tried to avoid loss of information and not to violate the assumption of independence by using “shifting unit of analysis”, which used an individual study as the unit of analysis when calculating the overall effect size. Specifically, when calculating the total effect size, this study averaged sub-factors to derive only one effect size from each study and used the mean value as the representative value of the study. However, when the focus of the analysis was on a sub-factor, the individual effect size within the study was used as the unit of analysis, not each study. This study used Comprehensive Meta-analysis (CMA) software version 2.0 for analyzing accurate effect sizes. CMA software is widely used worldwide because it can explain complex problems such as the effect of weights, the meaning of heterogeneity, and the difference between fixed and random effects.

## 3. Results

### 3.1. Flow Diagram for Search and Selection 

The database searched keywords from the beginning until December 2021 and found 20,480 studies. After removing duplicates, 8280 studies were collected for the abstract and title screening. Then, we conducted full-text screening for 102 screened studies. Finally, this study selected five studies relevant to hepatic disease for analysis. The flow diagram of literature selection progression is shown in Figure 1.

### 3.2. Characteristics of Included Studies

The characteristics of the five cohort studies (i.e., prospective cohort studies and retrospective cohort studies) are described in detail in Table 1. The subjects of the five selected cohort studies ranged from 67 to 32,254 people. These studies were conducted between 1993 and 2014 in the USA, Sweden, or Denmark. Among the five studies, a marker of liver function alanine transaminase (ALT) increased with PFOA exposure in two studies [15,16]. Specifically, gamma-glutamyl transferase (GGT) was positively associated with PFOA exposure [15], whereas the other study showed no evidence of GGT [16]. In the other three studies, PFOA exposure in the general population showed no association with the risk of liver cancer [17,18,19]. Table 1 and Figure 2 show the RoB of the five cohort studies.

### 3.3. Meta-Analysis of PFOA Exposure on Hepatic Disease

This study performed a meta-analysis using nine effect sizes derived from two studies for the association between PFOA exposure and hepatic disease. This study conducted the homogeneity test based on the assumption that the results of each study were from the same population to find that heterogeneity was not significant (Q(1) = 0.249, *p* > 0.05), indicating that there was no heterogeneity in the effect size between individual studies. Therefore, it was concluded that the results of the fixed effect model and those of the random effect model were the same (Appendix A). Three studies that were not included in the meta-analysis were excluded from the analysis because they did not have sufficient usable statistical results. This study calculated the whole effect size of the odds ratio for PFOA exposure studies using meta-analysis. The whole effect size was 1.169 and the confidence interval was between 1.089 and 1.256, which were significant. In other words, the analysis results showed that effect size values for each study of the PFOA exposure were significantly higher than that of the PFOA non-exposure. Appendix A shows the forest plot presenting the effect size of each study immediately. Figure 3 depicts the analysis results associated with PFOA exposure according to variables related to hepatic disease in this study. The heterogeneity between groups was Q(2) = 1.600 (*p* > 0.05), indicating that the difference in mean effect size between groups was not significant. The odds ratio of ALT was 1.167, and the confidence interval was from 1.086 to 1.254, which was significant. In other words, when it was exposed to PFOA, ALT significantly increased by 117% compared to no PFOA exposure. It was found that cancer and disease were not significant because the confidence interval included 1.

## 4. Discussion

It has been reported that exposure to PFOA would cause hepatotoxicity, hepatic disease, reproductive and developmental toxicity, and cardiovascular disease in humans, and immune and endocrine disorders in animals. However, there are relatively few studies on the effects of PFOA exposure on hepatotoxicity and hepatic disease than other diseases. Most of the meta-analysis studies related to PFOA exposure are regarding reproductive and developmental toxicity [20,21,22,23,24,25,26] and cardiovascular disease [27]. It is difficult to find a meta-analysis study on hepatotoxicity and hepatic disease.

It is not clear whether PFOA exposure is related to hepatotoxicity and hepatic disease, because previous epidemiological studies have reported conflicting results. This is the first meta-analysis study to provide evidence that PFOA exposure may affect the pathogenesis of hepatic injury. This study conducted a meta-analysis on the correlation between PFOA exposure and the development of hepatic disease in adults and showed that PFOA was positively correlated with alanine aminotransferase (ALT). The assay of the serum activity of the enzyme ALT is not an accurate measure of the severity of the hepatic injury or dysfunction. Nevertheless, ALT is the standard screening tool for detecting acute hepatic injury and is by far the most used [28,29]. The results imply that PFOA exposure may affect liver function and cause hepatic injury.

However, PFOA exposure and how it triggers liver function or hepatic injury is still unclear. A recently conducted experiment on Sprague Dawley rats reported that PFOA significantly changed ALT and induced liver edema and liver toxicity by using mechanisms related to specific protein denaturation [30]. The mechanism of PFOA’s hepatotoxicity was also studied by using the human liver cell line (HL-7702), and a relationship with specific protein denaturation was reported. However, it also could not reveal an exact mechanism related to hepatotoxicity in humans [31].

Consequently, studies that can help elucidate the mechanism of liver function and hepatic injury in humans have been published. The relationship between PFOA exposure and liver enzymes was examined using 9523 Americans aged 20 years or older extracted from the serum data of the National Health and Nutrition Examination Survey (NHANES), and it was presumed that the renal glomerular function could influence the changes in liver enzymes (e.g., ALT) caused by PFOA exposure [32]. The possible health risks caused by exposure to PFOA through ingestion of foods such as seafood, dairy products, and drinking water were examined, and it showed that PFOA is readily absorbed from the gastrointestinal tract and partially excreted through urine and feces [33]. It is believed that the results of these studies would help us understand the metabolic process of PFOA. Since PFOA is stable in the body and has a long half-life (2.3–8.5 years) [34], more in vivo and in vitro investigations are required to fully understand the mechanism of altered liver function and hepatic injury caused by PFOA exposure. However, recent studies suggest that the PFOA half-life in humans may be less than two years [35]. We believe that these studies will help us find out how PFOA exposure affects blood cholesterol levels and TG and non-alcoholic fatty liver disease, which has been additionally reported recently [7,36].

This study has several limitations. First, the number of studies included in the analysis is not large enough to clearly conclude that PFOA exposure is associated with developing the hepatic disease. Second, the number of studies included in this study was not large enough to divide the data into low-level PFOA exposure and high-level PFOA exposure or to examine the risk due to continued exposure. If more data are available, an analysis will become easier, and the conclusions will be more robust. Consequently, further studies need to include a population vulnerable to hepatic diseases. In addition, since people are exposed to multiple chemicals at the same time, they may be exposed to potential risks from simultaneous exposure to multiple chemicals or more risks from the mixture of chemicals. Therefore, additional studies are required to study the development of hepatic diseases due to the simultaneous exposure of mixtures of PFCs, as well as the effects of a single PFOA exposure.

## 5. Conclusions

The findings of this systematic review and meta-analysis suggest that the ALT of people exposed to PFOA was higher than those not exposed. These results imply that PFOA exposure may affect liver function and cause hepatic injury. However, since the number of studies included in the analysis was not large enough to clearly conclude that PFOA exposure was associated with the development of hepatic diseases, more observational studies and high-quality, multicentered studies are needed to confirm its long-term effects.

## Figures and Tables

**Figure 1 ijerph-19-11318-f001:**
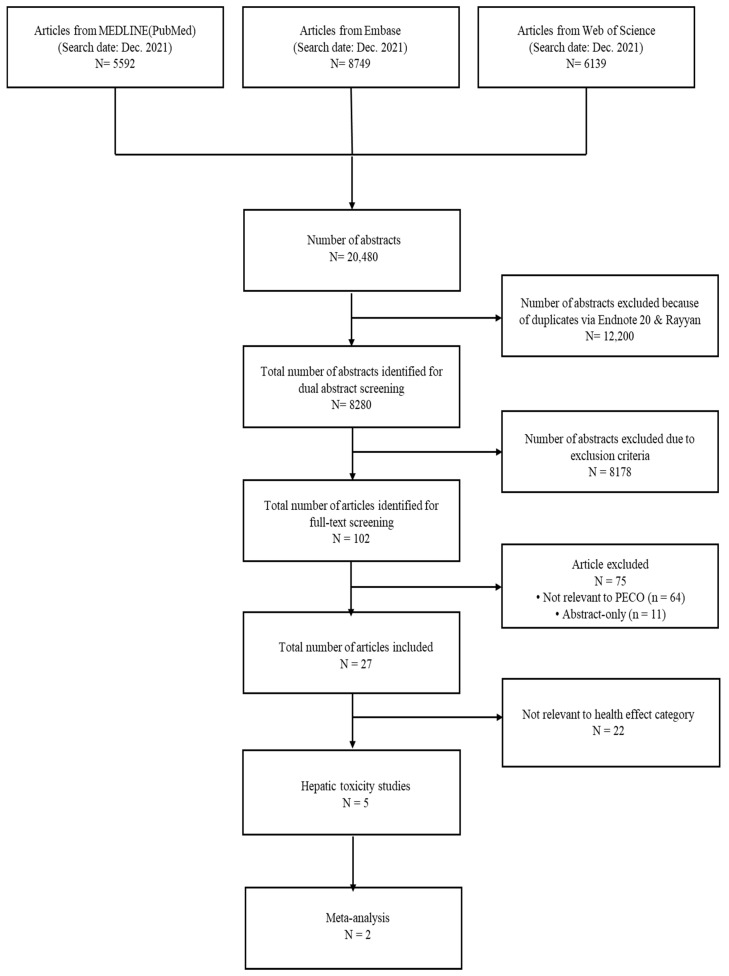
Flow diagram presenting the search and selection processes of the meta-analysis using studies on the exposure to PFOA in hepatic disease.

**Figure 2 ijerph-19-11318-f002:**
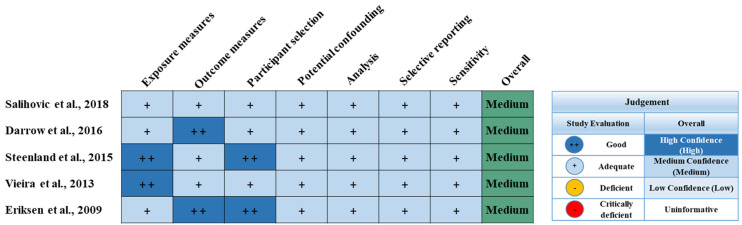
Summary of the risk of bias judgments for each included epidemiology study [15,16,17,18,19].

**Figure 3 ijerph-19-11318-f003:**
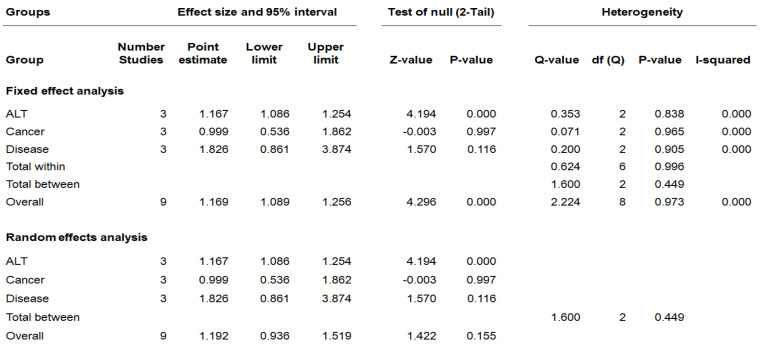
Effect size of PFOA exposure according to variables related to hepatic disease.

**Table 1 ijerph-19-11318-t001:** Characteristics and risk of bias assessment of the epidemiology studies.

Outcome Categories	Author, Year	Study Design	Location (e.g., Country)	Study Period	Population			Exposure	Sample matrix	Outcome	Mean Exposure (IQR)	Effect Estimate	Study Evaluation							
					Participant	N Enrolled	Mean (SD) Age, yr						Exposure	Outcome	Selection	Confounding	Analysis	Reporting	Sensitivity	Overall confidenc
Hepatic	Salihovic et al., 2018 [15]	Longitudinal study	Sweden	2001–2014	General population, elderly (age 70: 2001–2004) (age 75: 2006–2009) (age 80: 2011–2014)	1002 (age 70), 817 (age 75), 603 (age 80)	70.0 ± 0.2	PFOA	Plasma	Bilirubin, μmol/L PFOA −1.39 (−1.78, −1.01)	PFOA (median (IQR)) Age 70:3.31 (2.52, 4.39); Age 75:3.81 (2.71, 5.41); Age 80:2.53 (1.82, 3.61)	Coefficients (β) (CI)	A	A	A	A	A	A	A	Medium
										ALT, μkat/L PFOA 0.04 (0.03, 0.06)										
										ALP, μkat/L PFOA 0.11 (0.06, 0.15)										
										GGT, μkat/L PFOA 0.07 (0.01, 0.12)										
Hepatic	Darrow et al., 2016 [16]	Cohort study (retrospective)	USA	2005-2011	C8 Health Project Survey Participants August 2005–August 2006 and Original DuPont Cohort	Liver biomarkers n = 30,723 (including 1892 workers), combined cohort for liver disease n = 32,254	46 (for liver disease)	PFOA	Serum	OR (95% CI) for above normal ALT 1.04 (1.01,1.07) Q1: reference; Q2: 1.12 (1.00, 1.27); Q3: 1.14 (1.01, 1.29); Q4: 1.20 (1.06, 1.35); Q5: 1.16 (1.02, 1.33) (p for trend: 0.0078)	Q1: 50.3–191.2; Q2: 191.2–311.3; Q3: 311.3–794.1; Q4: 794.1–3997.6; Q5: 3997.6–20,5667.3	OR (CI)	A	G	A	A	A	A	A	Medium
Hepatic	Steenland et al., 2015 [17]	Cohort study (retrospective)	USA	2008–2011	The C8 Health Project (C8HP), workers employed between 1948 and 2002 at a DuPont plant in West Virginia	3713	Mean birth year: 1951 (SD 14)	PFOA	Serum	Non-hepatitis liver disease (10-years lag) Q1 1.00, Q2 1.46 (0.42, 5.04), Q3 2.13 (0.59, 7.71), Q4 2.02 (0.50, 8.10)	Mean measured exposure: 325 (SD 920) Mean predicted exposure: 218 (SD 358)	RR (CI)	G	A	G	A	A	A	A	Medium
Hepatic	Vieira et al., 2013 [18]	Cohort study (retrospective)	USA	1996–2005	OH Cancer Incidence Surveillance System (OCISS), WV Cancer Registry (WVCR)	25,107 (7869 OH cases and 17,238 WV cases)	Median 67 years	PFOA	Serum	Liver cancer 0.9 (0.3, 2.5)	Medium = 12.9–30.7µg/L (very high = 110–655 µg/L; high = 30.8–109 µg/L; medium = 12.9–30.7 µg/L; low = 3.7–12.8 µg/L)	AOR (CI)	G	A	A	A	A	A	A	Medium
Hepatic	Eriksen et al., 2009 [19]	Cohort study (prospective)	Denmark	1993–2006	Diet, Cancer and Health (DCH) cohort, Patients with prostate, bladder, pancreatic, liver cancer	1240 (prostate cancer n = 713, liver cancer n = 67)	Median 59 years (5–95th percentiles: 51–65)	PFOA	Plasma	PFOA (quartile trend) Liver cancer 0.95 (0.86, 1.06)	Liver cancer PFOA median 5.4 (5–95th percentiles: 2.5–13.7)	RR (CI)	A	G	G	A	A	A	A	Medium

IQR = interquartile range, G = good; A = adequate; D = deficient; C = critically deficient; high = high confidence; medium = medium confidence; low = low confidence.

## Data Availability

The original contributions presented in the study are included in the article/Appendix A, and further inquiries can be directed to the corresponding author/s.

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
