# Peer review of "Human Evidence of Perfluorooctanoic Acid (PFOA) Exposure on Hepatic Disease: A Systematic Review and Meta-Analysis"

_ijerph, 2022, doi:10.3390/ijerph191811318_

Round 1
Reviewer 1 Report
1. Is PFOA considered as Persistent Organic Pollutants until 2020? Exists a previous report in the literature?.
2. Figure 1, Excluded (n=75): is “Other (n=75)” repeated?
3. The resolution of Supplementary Figure S1 is too low and the data is unclear.
4. line 226-231 “……hepatotoxicity or liver diseases is still unclear.”: This passage is more appropriate for the introduction section.
5. line 248: the half-life of PFOA was reported as 2.3-8.5 years (Wang Z, Cousins IT, Scheringer M et al. (2013) Fluorinated alternatives to long-chain perfluoroalkyl carboxylic acids (PFCAs), perfluoroalkane sulfonic acids (PFSAs) and their potential precursors. Environment International 60:242-248).
6. Why is not the study “Girardi P, Merler E (2019) A mortality study on male subjects exposed to polyfluoroalkyl acids with high internal dose of perfluorooctanoic acid. Environmental Research 179:108743“ included?
7. line 267-268: this is not a concise enough sentence.
Reviewer 2 Report
Substantive Comments:
The overall conclusions by the authors are correct. Their study has severe limitations that do not allow any conclusions of the effect of PFOA on liver function. The several studies listed in Table 1 would actually belie this effect of PFOA. I suggest that the authors reanalyze their data using a systematic review that is not as intense as suggested by EPA so that additional studies can be included in the meta-analysis. Otherwise, the authors should wait for additional findings so that a more definitive conclusion is possible.
Marginal Comments:
Lines 222-225: “This study conducted a meta-analysis on the correlation between PFOA exposure and the development of hepatic disease in adults and showed that PFOA was positively correlated with Alanine Aminotransferease (ALT). The results imply that PFOA exposure may affect liver function and cause hepatic disease.”
Comment: And yet other indicators of liver disease in several of the studies listed in Table 1 involving thousands of people did not show liver disease. The authors should comment on this disparity.
Lines 230-231: “However, how it triggers hepatotoxicity or liver diseases is still unclear.
Comment: But liver disease has not been demonstrated in this study. Only one marker of liver function is impacted, and this is apparently in only one of two studies.
Lines 242-243: “…confirmed that the renal glomerular function could influence the changes in liver enzymes caused by PFOA exposure [30].”
Comment: This sentence implies a causal relationship whereas the title of the cited reference suggests only an association.
Round 2
Reviewer 2 Report
I appreciate the authors attempt to resuscitate a manuscript that otherwise would not be publishable, through no fault of their own apparently. The available literature does not appear to be sufficient to conduct a rigorous meta-analysis. The authors appear to have done the best possible job, although waiting for additional studies, or integrating their minimal findings with experimental animal work might also have been attempted. My only further suggestion is to consider more recent findings that the half-life of PFOA may not be the as long as they reference. See for example, Campbell, Jerry, Harvey Clewell, Tony Cox, Michael Dourson, Shannon Ethridge, Norman, Forsberg, Bernard Gadagbui, Ali Hamade, Ravi Naidu, Nathan Pechacek, Tiago Severo Peixe, Robyn Prueitt, Mahesh Rachamalla, Lorenz Rhomberg, James Smith, Nitin Verma. 2022. The Conundrum of the PFOA human half-life, an international collaboration. Regulatory Toxicology and Pharmacology 132 (2022) 105185.
